# Cellular Senescence: A Troy Horse in Pulmonary Fibrosis

**DOI:** 10.3390/ijms242216410

**Published:** 2023-11-16

**Authors:** Ruyan Wan, Lan Wang, Miaomiao Zhu, Wenwen Li, Yudi Duan, Guoying Yu

**Affiliations:** 1Henan International Joint Laboratory of Pulmonary Fibrosis, Henan Center for Outstanding Overseas Scientists of Pulmonary Fibrosis, College of Life Science, Institute of Biomedical Science, Henan Normal University, Xinxiang 453007, China; 1904084050@stu.htu.edu.cn (R.W.); 041099@htu.cn (L.W.); 2104183072@stu.htu.edu.cn (M.Z.); 2104183010@stu.htu.edu.cn (W.L.); 2104183035@stu.htu.edu.cn (Y.D.); 2State Key Laboratory Cell Differentiation and Regulation, Henan Normal University, Xinxiang 453007, China

**Keywords:** cellular senescence, pulmonary fibrosis, pathogenesis, treatment

## Abstract

Pulmonary fibrosis (PF) is a chronic interstitial lung disease characterized by myofibroblast abnormal activation and extracellular matrix deposition. However, the pathogenesis of PF remains unclear, and treatment options are limited. Epidemiological studies have shown that the average age of PF patients is estimated to be over 65 years, and the incidence of the disease increases with age. Therefore, PF is considered an age-related disease. A preliminary study on PF patients demonstrated that the combination therapy of the anti-senescence drugs dasatinib and quercetin improved physical functional indicators. Given the global aging population and the role of cellular senescence in tissue and organ aging, understanding the impact of cellular senescence on PF is of growing interest. This article systematically summarizes the causes and signaling pathways of cellular senescence in PF. It also objectively analyzes the impact of senescence in AECs and fibroblasts on PF development. Furthermore, potential intervention methods targeting cellular senescence in PF treatment are discussed. This review not only provides a strong theoretical foundation for understanding and manipulating cellular senescence, developing new therapies to improve age-related diseases, and extending a healthy lifespan but also offers hope for reversing the toxicity caused by the massive accumulation of senescence cells in humans.

## 1. Introduction

Pulmonary fibrosis (PF) is a chronic, progressive, devastating, and irreversible interstitial lung disease, with a median survival of 2 to 3 years after diagnosis [1,2]. The present comprehension of the pathogenesis of PF entails the repetitive injury of alveolar epithelial cells (AECs) due to various risk factors, such as environmental exposure, viral infections, genetic predisposition, oxidative stress, and immunological factors [3,4]. This injury subsequently results in the abnormal activation of AECs and dysregulated epithelial repair processes [3,4]. The dysregulated epithelial cell secretes multiple cytokines and growth factors and interacts with endothelial, mesenchymal, and immune cells via multiple signaling mechanisms to trigger fibroblast and myofibroblast activation and promote extracellular matrix deposition, ultimately leading to the destruction of lung function, diminished exercise tolerance, and a decreased quality of life [4,5]. The existing epidemiological data from various data sources indicate that the average age of patients with PF is estimated to be over 65 years, and the incidence increases with age [6]. Furthermore, individuals aged 70 and above have a risk of developing PF that is seven times higher than those in their 40 s. Therefore, PF is now considered an age-related lung disease [7]. Among the hallmarks of aging, cellular senescence serves as the primary driver behind tissue and organ aging, as well as an independent risk factor for PF progression. Age-related disturbances were increasingly observed in epithelial cells and fibroblasts in PF lungs compared to age-matched cells in normal lungs. Physiologically, alveolar epithelial type II (ATII) cells, serving as progenitor cells of the alveoli, differentiate into ATI cells in response to injury. Utilizing organoid cultures, single-cell transcriptomics, and lineage tracing, it has been discovered that ATII cells differentiate into ATI cells and acquire a transitional state known as pre-alveolar type 1 cell during the process of maturation. This transitional state exhibits regulation by TP53 signaling, making it susceptible to DNA damage and undergoing transient senescence [8]. However, there are at least two harmful consequences of persistent senescence. On the one hand, telomere wear and mitochondrial dysfunction lead to permanent cell-cycle arrest, which in turn causes stem cell/progenitor cell-renewal dysfunction and the loss of self-repair and regeneration abilities. On the other hand, senescent cells produce pro-inflammatory, pro-fibrotic, and stroma-remodeling cytokines such as IL-6, TGF-β, and several matrix metalloproteinases collectively known as the senescence-associated secretory phenotype (SASP), which can activate myofibroblast and scar formation. In fact, some components of SASP appear to enhance the growth arrest of exposed adjacent cells in a paracrine manner, further driving senescence, leading to low-grade chronic inflammation, and increasing susceptibility to pulmonary fibrosis [9,10,11]. Given the aging global population, the psychological, physical, and socio-economic burdens associated with PF is substantial [12]. A comprehensive understanding of how senescence promotes the occurrence and progression of PF can provide new insights into the further treatment of age-related diseases. This study presents compelling new evidence indicating that cellular senescence is a significant driving factor in age-related lung diseases such as PF. It not only systematically summarizes the causes of cellular senescence in PF and the signaling pathways regulating different types of cellular senescence but also provides potential therapeutic strategies for targeting cellular senescence to improve PF. These strategies include targeting the clearance of senescent cells, intervening in senescence-related signaling pathways, and inhibiting the secretion of SASP.

## 2. Perspective on Cellular Senescence in Pulmonary Fibrosis

### 2.1. Cellular Senescence in Pulmonary Fibrosis

The world’s population is aging at an unprecedented rate in human history. It is projected that by 2050, the number of individuals aged 65 and above will surpass 2 billion worldwide, constituting approximately 20 percent of the global population [13]. Aging serves as a significant contributor to the mortality associated with age-related chronic diseases in adults. It is crucial to acknowledge that advanced age alone hampers the physiological functioning of the lungs, even in healthy persons. In general people, lung function reaches its maximum around the age of 25 for men and 20 for women, after which it gradually diminishes with age. Specifically, as a person ages, there is a progressive loss of the alveolar surface area and a decrease in ciliary mucus clearance and the ratio of forced expiratory volume in one second (FEV1)/forced vital capacity (FVC), and there are changes in the elasticity of the lungs [14]. Aged lungs may also be more vulnerable to damage caused by environmental exposures, such as aerosols in the air, cigarette smoke, particulates from diesel emissions, and other exogenous toxins, and this may be due to changes in lung function and anatomy, lung antioxidant defenses, and innate and adaptive immune responses [15].

Cellular senescence, a hallmark of aging, is defined as irreversible cell-cycle arrest caused by various intrinsic and extrinsic stimuli, such as DNA damage, oncogene activation, telomere shortening, mitochondrial dysfunction, and mechanical stress that alter cellular morphology and transcription, secrete a range of unique pro-inflammatory and proteolytic factors, and modulate the local tissue microenvironment through autocrine and paracrine signaling [16,17]. Since the 1960s, when Hayflick and Moorhead first observed the limited replication potential of normally cultured human lung fibroblasts, numerous stimuli have been associated with the accumulation of senescent cells within the pulmonary system. It has been found that senescent cells accumulate in the epithelial cells, endothelial cells, fibroblasts, and immune compartments of the human lung with age [18]. In fact, all the “hallmarks of aging”, including genomic instability, altered cellular communication, telomere attrition, stem cell exhaustion, epigenetic changes, cellular senescence, loss of proteostasis, mitochondrial dysfunction, and dysregulated nutrient sensing, were observed in PF patients, suggesting that the pathobiology of PF is closely related to aging [19]. Emerging single-cell RNA sequencing has been widely applied to depict the cellular composition and functional phenotypes in fibrotic lung tissue [20,21,22]. Multiple studies have demonstrated that various cell types, including endothelial cells, epithelial cells, and fibroblasts, exhibit a senescence-like phenotype in the lung tissue of fibrotic patients. Yao et al. performed a single-cell RNA sequencing analysis and fluorescence-activated cell sorting on epithelial cells from proximal and distal lung tissues of nine control donors and 11 IPF fibrotic lung tissues to assess the contribution of epithelial cells to the senescent cell fraction in IPF [23]. The results revealed that ATII cells isolated from IPF patient lung tissues exhibited characteristic transcriptomic features of cellular senescence, and the senescence of ATII cells, rather than their loss, was sufficient to drive progressive lung fibrosis [23]. Alveolar macrophages are resident innate immune cells that play a crucial role in maintaining lung physiological functions. Wu et al. revealed, through single-cell RNA sequencing, that the potential mechanism underlying the aging-like phenotype of alveolar macrophages is attributed to the loss of transcription factor CBFβ activity [24]. In radiation or immune therapy-induced lung injuries, single-cell RNA sequencing analysis has revealed that the accumulation of senescence-like fibroblasts, alveolar epithelial cells, and macrophages is a primary pathological mechanism [25]. In addition, a time-series single-cell RNA-seq analysis using a bleomycin-induced pulmonary injury model revealed the persistent presence of Krt8+ transitional stem cells exhibiting transcriptional features of cellular senescence during the process of alveolar regeneration in human pulmonary fibrosis [26]. Further experimental validation can also detect the expression of cellular-senescence markers in AECs and lung fibroblasts of PF patients. Senescent lung fibroblasts exhibit an excessive secretion of extracellular matrix components, thereby contributing to the progression of PF [23,27,28]. Importantly, the selective removal of senescent cells rejuvenates pulmonary health and alleviates PF in aged mice [29].

### 2.2. Inducer of Cellular Senescence in Pulmonary Fibrosis

#### 2.2.1. Oxidative Stress

Oxidative stress caused by an imbalance between oxidant production and the antioxidant defense system and/or the overproduction of ROS can induce cell senescence [30]. A variety of factors, including mitochondrial dysfunction, endoplasmic reticulum stress, and radiation, can interrupt cell redox homeostasis, thus inducing DNA damage and leading to cell senescence. For example, the persistent elevation of mitochondrial uncoupling protein-2 (UCP2) is closely related to an increased ROS production, an altered redox status, and the induction of fibroblast senescence. The inhibition of UCP2 expression can reduce ROS production in IPF lung fibroblasts, thereby alleviating cell senescence of lung fibroblasts and promoting the resolution of bleomycin-induced experimental PF [31]. It was found that Bmi-1-deficient mice induced cell senescence by ROS accumulation and DNA damage due to impaired mitochondrial functions and a redox imbalance. And TGF-β1/IL-11/MEK/ERK signaling increased senescence-related pulmonary fibrosis in Bmi-1 deficient mice [32]. In addition, the mitochondrial biogenesis pathway driven by the mTOR/PGC-1α/β axis was significantly increased in senescence AECs. mTORC1 complex pharmacological inhibition not only restores mitochondrial homeostasis but also reduces bleomycin-induced lung epithelial cell senescence [33]. As a key protein of the endoplasmic reticulum stress signaling pathway, the CCAAT/enhancer-binding protein (C/EBP) homologous protein (CHOP) activates the oxidative stress signaling pathway by promoting the generation of ROS, thus accelerating the senescence of AECs and PF [34]. Briefly, numerous factors regulate the process of cellular senescence by modulating the cellular redox status, thereby participating in the occurrence and progression of PF.

#### 2.2.2. DNA Damage

Most of the pathogenesis of PF induced by genetic and environmental factors, to a certain extent, will lead to DNA damage, such as DNA breakage, DNA oxidative damage, and so on [35]. For example, bleomycin, a commonly used chemotherapy medication, has been extensively employed to establish a rodent model of PF. The mechanism by which bleomycin-induced PF is commonly believed to be is that bleomycin induces DNA’s single- or double-strand breaks, resulting in an increased production of free radicals, which leads to lung inflammation and subsequent fibrosis [36]. Furthermore, multiple studies have shown that DNA damage is an inducer of cell senescence [37,38]. In a diabetes-induced PF model, there was an escalation in DNA damage and impairment in DNA repair. Sustained DNA damage induced cellular senescence and the secretion of SASP factors, such as pro-inflammatory cytokines and growth factors, ultimately leading to fibrosis [39].

#### 2.2.3. Telomere Attrition

Telomeres are short DNA repeat sequences found at the ends of linear chromosomes in eukaryotes that maintain chromosome integrity and control cell-division cycles, playing a crucial role in cell fate and aging. The homeostasis of telomere length is crucial for proper cell function. However, due to the mechanism of replication, cell proliferation leads to telomere shortening, which hinders DNA replication and activates the DNA damage response, leading to cellular replicative senescence and mitochondrial dysfunction [40]. Telomerase, an enzyme responsible for counteracting telomere shortening during cell division, has been identified with mutations in various diseases, including PF [41]. In fact, up to 15% of familial PF cases have rare mutations in the human telomerase reverse transcriptase (TERT) gene, which determines cell senescence by regulating telomere length [42]. In addition, the AECII-specific lack of the TERT gene enhances PF by increasing susceptibility to bleomycin-induced epithelial injury and cellular senescence [43]. Under the stimulation of oxidative stress, radiation, or bleomycin, the E3 ubiquitin ligase FBW7 accelerates the polyubiquitination and degradation of telomere protection protein 1 (TPP1) by binding to TPP1, leading to telomere shortening and inducing ATII cell senescence and PF [44].

#### 2.2.4. Oncogene Activation

In addition to telomere attrition, oxidative stress, DNA damage, and other factors that induce PF, oncogene activation can also induce PF accompanied by cell senescence. The c-Src protein, encoded by the proto-oncogene SRC, as a nonreceptor tyrosine kinase, has been shown to promote the development of organ fibrosis. It was found that c-Src activity was elevated in a silica-induced PF mouse model, and the inhibition of Src kinase reduced myofibroblast activation and alleviated the progression of PF [45]. Gao AY et al. discovered that Pim-1 kinase promotes the production of IL-6 by activating NF-κB activity, leading to the manifestation of a senescent phenotype in low-passage IPF fibroblasts. Furthermore, the targeted inhibition of Pim-1 kinase effectively suppresses the secretion of SASP in senescent fibroblasts and halts the progression of IPF [46]. Additionally, RAS activation results in growth arrest and the increased SASP secretion of human lung fibroblasts [47]. These studies suggest that oncogene activation is a positive feedback regulator of cell senescence and PF.

#### 2.2.5. Ionizing Radiation

Thoracic radiation therapy is the fundamental treatment for non-small cell lung cancer, esophageal cancer, breast cancer, and various mediastinal tumors. However, a range of microenvironmental stress responses, including cellular senescence, are triggered after radiation therapy [48]. Accumulating evidence suggests that ionizing radiation (IR)-induced cell senescence may be a driving factor in the pathogenesis of radiation-induced pulmonary fibrosis (RIPF) [48,49,50]. Cancer-associated fibroblasts were isolated from surgically resected lung cancer specimens of patients with non-small cell lung cancer undergoing radiotherapy and showed senescence-like characteristics after a single dose of 10 gray radiation treatments. In vivo, targeting senescence-like fibroblasts with FOXO4-p53-interfering peptide FOXO4-DR alleviates RIPF [50].

In addition, the senescence-associated β-galactosidase (SA-β-Gal) activity, the expression of senescence-specific genes (p16, p21), senescence-associated secretory phenotype (SASP) chemokines and proinflammatory (Ccl2, Ccl17, Cxcl10, Il-1α, and Il-6α) and pro-fibrotic factors (TGF-β1 and Arg-11) were all increased in bone marrow-derived monocytes/macrophages under 10 gray IR. This suggests that IR can induce the senescence of macrophages and thus stimulate the fibrotic phenotype of lung fibroblasts [51]. The mechanism underlying cell senescence induced by radiation is believed to involve the detrimental effects of ionization and/or reactions with free radicals on the integrity and functionality of DNA, proteins, and lipids. These damages subsequently trigger metabolic and functional alterations, ultimately culminating in cellular senescence [48].

## 3. Main Signaling Pathways Governing Cellular Senescence in Pulmonary Fibrosis

### 3.1. Cell-Cycle Arrest Regulatory Pathway

Irreversible cell-cycle arrest due to DNA damage, oncogene activation, mitochondrial dysfunction, oxidative stress, and other harmful stimuli or abnormal proliferation is a common feature of senescent cells. Senescent cells still have cell viability and metabolic activity. The cell-cycle arrest caused by senescent cells differs from that induced by quiescent and terminally differentiated cells [52]. Quiescence is a state in which cell growth is temporarily arrested due to contact inhibition, nutrient deprivation, and other processes but can continue to proliferate when growth conditions are restored. Terminal differentiation is not a random cell response to injury but is rather developmentally specified, with morphological and functional changes. Senescent cells can acquire new phenotypes. In mammalian cells, two interacting but independent signaling pathways, p53-p21CIP1 and p16Ink4a-Retinoblastoma (Rb), play a vital role in establishing cell-cycle arrest during senescence [53]. When the DNA double-strand breaks, the cell will initiate the DNA damage response (DDR). For example, in the case of DNA damage induced by IR, p53 activates DDR signaling to facilitate DNA repair. However, if the DNA damage is beyond repair, p53 induces cell senescence, halting the progression of the cell cycle [54]. The upregulation of p53 activity in primary ATII cells isolated from the lung tissues of a stress-induced premature senescence mouse model with Bmi-1 deficiency diminished their capacity to proliferate and differentiate, thereby promoting the senescence phenotype and the progression of PF [32]. In addition, the expression level of p53 was significantly elevated in a bleomycin- and radiation-induced PF mouse model and the ATII cells of IPF explant tissue [55,56,57]. However, the inhibition of p53 signaling alleviates lung fibrosis caused by Sin3a loss of function in ATII cells [23]. Moreover, when the senolytic cocktail of dasatinib plus quercetin was used to improve the PF in mice, the expression of p53 was significantly reduced [23]. These findings indicate that p53 can promote the senescence of lung cells and thus aggravate the progression of PF.

At the same time, p53 also regulates the expression of the cell cycle-dependent kinase inhibitor p21CIP1, which induces cell-cycle arrest and cell senescence [58]. The p21 protein interacts with other proteins involved in cell-cycle regulation, such as cyclin/CDK, PCNA, etc., through two conserved domains, thereby inhibiting the phosphorylation of Rb and subsequently arresting the cell cycle [59]. A growing body of evidence implicates that the expression of p21 was significantly upregulated in the lung tissues of PF mice induced by radiation and single- or multiple-dose bleomycin, and the senescence of primary ATII cells isolated from it increased [60,61,62]. However, in mice with Grp78 or Sin3a deficiency-induced PF, p21 and p53 expression were significantly reduced after treatment with dasatinib plus quercetin [23,57]. In addition, the bubble-like nanoparticles formed by encapsulating Arctiin using 1,2-distearoyl-sn-glycero-3-phosphoethanolamine and polyethylene glycol 2000 as carriers suppress ATII senescence and alleviate bleomycin-induced PF by inhibiting the p53/p21 signaling pathway [63]. In conclusion, the cell-cycle inhibitor p21 promotes the occurrence of PF by regulating the cell cycle and influencing the cell state.

Rb, a central regulator of the cell cycle in mammalian cells, mainly acts as a transcriptional suppressor. It was found that the Rb protein forms complexes with transcription factors of the E2F family and down-regulates the expression of many cell-cycle regulators. Conversely, the phosphorylation of Rb represses the transcription of the RB-E2F complex, thereby up-regulating the expression of cell-cycle regulators and promoting cell proliferation [64]. However, in senescent cells, the accumulation of the CDK4/6 inhibitor p16INK4A prevents the phosphorylation of Rb, leading to G1 cell-cycle arrest [65]. Notably, accumulating evidence indicates that an increased p16 expression appears to drive the progression of PF by various mechanisms [51,66,67]. In an RIPF model, bone marrow-derived monocyte/macrophage (BMM) cells showed a senescent phenotype after IR, and the expression of p16INK4a increased with the degree of senescence of BMM cells [51]. Meanwhile, TGF-β1 induced the senescence of primary mouse ATII cells and stimulated the secretion of pro-fibrotic mediators by activating the p16–pRb pathway [68]. The treatment of fibrotic primary mouse ATII cells and/or three-dimensional lung tissue cultures with dasatinib plus quercetin decreased senescent markers, including p16 and p21, and attenuated experimental PF. Collectively, these data provide compelling evidence for a correlation between the level of p16INK4a and the severity of fibrosis.

### 3.2. Senescence-Associated Secretory Phenotype (SASP) Regulatory Pathway

Cellular senescence is not merely an irreversible arrest of the cell cycle but also encompasses sustained cell viability and metabolic activity. Senescent cells undergo significant alterations in protein expression and secretion, ultimately giving rise to SASP. Substantial evidence suggests that SASP is widely present in diverse senescent cell populations found in the lungs of patients with PF and mouse models [46,69,70]. Senescent cells regulate the cellular and local tissue microenvironment in an autocrine and paracrine manner by secreting various forms of SASP components, including cytokines, chemokines, growth factors, extracellular matrix proteases, bioactive lipids, and noncoding nucleotides [71]. It has been found that SASP is regulated by the activation of specific transcription factors such as the nuclear factor kappa-light-chain enhancer of activated B cells (NF-κB), the CCAAT/enhancer-binding protein (C/EBP) and p53, chromatin remodeling, and the control of intracellular trafficking [72,73,74]. For example, in pulmonary fibrotic diseases, TGF-β acts as a prominent SASP factor, inducing and sustaining senescence phenotype and pathological conditions through autocrine/paracrine mechanisms. On the one hand, TGF-β induces senescence in human bronchial epithelial cells (HBECs) by upregulating the expression of the cyclin-dependent kinase inhibitor p21. On the other hand, TGF-β-induced senescence in HBECs leads to an increased secretion of IL-1β, which subsequently enhances myofibroblast activation and promotes PF [42]. In addition, the TGF-β signaling pathway can mediate the senescence of AECs by suppressing the expression of the TERT gene, thereby promoting PF [75]. TGF-β1 induces cellular senescence in ATII and stimulates the secretion of various cytokines and chemokines, such as IL-4 and IL-13, which subsequently activate alveolar macrophages, thereby contributing to the onset and progression of PF [68]. Conversely, enhanced BMP4 signaling mitigates the senescence of lung fibroblasts, consequently inhibiting PF by reducing the expression of TGF-β1 [76]. In conclusion, distinct forms of SASP not only serve as indicators of senescence during PF development but also actively participate in the senescence process and regulate the progression of PF.

### 3.3. cGAS–STING Regulatory Pathway

DNA damage response is a vital step leading to cellular senescence in the pathogenesis of PF, which is characterized by the expression of SASP including pro-inflammatory cytokines and growth factors. Cyclic GMP–AMP synthase (cGAS) is a cytoplasmic DNA sensor that can trigger a type of interferon pathway to activate innate immunity and plays a crucial role in cell senescence. Upon the detection of pathogenic DNA, cGAS generates a second messenger called cyclic GMP–AMP (cGAMP), which subsequently activates the adaptor protein STING. Subsequently, STING recruits TANK-binding kinase 1 (TBK1) and i-κB kinase (IKK), leading to the activation of IFN regulatory factor 3 (IRF3) and NF-κB, respectively. This activation induces the expression of type I interferons (IFNs) and other inflammatory cytokines and chemokines [77]. A recent study has also discovered that the cGAS–STING pathway plays a vital role in the damage-induced senescence of ATII cells and the development of pulmonary fibrotic disease [78,79]. In vitro culture experiments demonstrated that the pharmacological inhibition or genetic knockdown of cGAS reduced DNA damage-induced senescence in primary AECs obtained from healthy donors and the attenuated senescence in primary AECs isolated from IPF patients [79]. Moreover, in models of PM2.5-induced lung cell senescence and lung tissue injury, pretreatment with selenomethionine inhibited the inflammatory response by blocking the cGAS/STING/NF-κB pathway, thereby alleviating senescence in lung cells and tissues [80]. Indeed, the pro-senescence cGAS–STING signaling pathway facilitates the synthesis and release of SASP components. Conversely, the lack of cGAS abolishes the expression of the cellular markers of senescence and genes related to SASP, including IL6, IL8, and MMP12 [81]. So, activation of the cGAS–STING signaling pathway plays a crucial role in driving PF progression, and targeting this pathway holds promise for PF treatment.

### 3.4. Wnt/β-Catenin Regulatory Pathway

The Wnt/β-catenin pathway is an evolutionarily conserved signaling system that is thought to play a critical role in regulating embryonic development and tissue homeostasis. At the cellular level, Wnt/β-catenin signaling regulates cell proliferation and viability, cell polarity establishment, and the self-renewal of stem cells. This pathway consists of the Wnt protein family of the extracellular signal, the Wnt receptor Frizzled and LRP5/6 of the cell membrane segment, β-catenin, DVL, GSK-3β, AXIN, APC, and CK1 of the cytoplasmic segment, and the TCF/LEF family members and downstream target genes of β-catenin of the nuclear segment [82]. Studies have shown the significant activation of Wnt/β-catenin signaling in various cell types of both human and experimental PF, suggesting that it may be a new therapeutic target in PF disease [83,84]. Lehmann et al. observed an elevated Wnt/β-catenin activity in ATII cells isolated from aged mice compared to those from young mice. Moreover, the chronic activation of canonical Wnt/β-catenin signaling induced a pronounced cellular senescence phenotype in primary ATII cells and MLE-12 cells [85]. Kadota T et al. reported that human bronchial epithelial-derived extracellular vesicles alleviated TGF-β-induced myofibroblast differentiation and alveolar epithelial cell senescence by inhibiting Wnt signaling pathways, thereby attenuating bleomycin-induced PF in mice [86]. Meanwhile, the induction of senescence in AECs by mitomycin was observed to be regulated through the enhancement of GSK3β phosphorylation [87]. Notably, the activation of the β-catenin signaling pathway initiates a DNA damage response, leading to the upregulation of senescence-related genes such as p16, p21, and p53 [88,89]. Consequently, the aberrant activation of Wnt/β-catenin signaling can promote lung cell senescence by inducing cell-cycle arrest, thereby contributing to the progression of PF.

### 3.5. IGF-1 Regulatory Pathway

Cell senescence is characterized by the arrest of cell-cycle progression, the enlargement and flattening of cell morphology, and the maintenance of cell viability and metabolic activity. It has been found that insulin-like growth factor 1 (IGF-1), a natural growth hormone, plays a crucial role in regulating cell growth, division, proliferation, and senescence [90,91]. In both human embryonic lung fibroblasts and mouse embryonic fibroblasts, prolonged exposure to IGF-1 resulted in premature cellular senescence by inhibiting the activity of SIRT1 deacetylase and enhancing the stability of p53 [91]. The inhibition or loss of the IGF-1 receptor has been shown to extend the lifespan of organisms by preventing cellular and tissue senescence [92,93]. In addition, the expression and concentration of IGF-1 were significantly increased in IPF patients and mouse models of bleomycin and radiation-induced PF, suggesting that IGF-1 is closely related to the development of PF [91,94]. Recent studies have also revealed that IGF-1 can induce AEC senescence and aggravate bleomycin-induced PF by activating the PI3K/AKT signaling pathway and augmenting the release of CTGF, MMP9, and TGF-β1 [91]. Mice lacking the IGF-1 receptor exhibited significantly reduced levels of AECII senescence and M2 macrophage accumulation, consequently alleviating PF [94].

### 3.6. NF-κB Regulatory Pathway

NF-κB is a transcription factor activated by cells in response to injury, stress, and inflammation, and its activity increases with aging and age-related chronic diseases [46,95,96]. In mammals, the NF-κB family comprises five subunits: RelA (p65), RelB, c-Rel, NF-κB1 (p50), and NF-κB2 (p52). The inhibition of NF-κB signaling components, such as p65, has been proven to inhibit SASP, including pro-inflammatory factors IL-6 and IL-8, thereby delaying cell senescence [97]. Accumulating evidence has indicated a significant elevation in the phosphorylation level of NF-κB in both bleomycin-induced mouse lung tissues and IPF patient lung tissues [98,99]. Tian Y et al. reported that the knockdown of PTEN accelerated bleomycin-induced cellular senescence and increased NF-κB activation in A549 cell lines and rat primary ATII cells. Conversely, the pharmacological inhibition or knockdown of NF-κB reversed bleomycin-induced alveolar epithelial cell senescence. These findings indicate that the deletion of PTEN accelerates AEC senescence by activating NF-κB signaling [100]. Additionally, the activation of the NF-κB signaling pathway by the CCAAT/enhancer-binding protein (C/EBP) homologous protein (CHOP) contributes to the increased senescence of AECs and PF induced by ER stress [34]. NF-κB not only regulates AEC senescence but also promotes the senescence of lung fibroblasts. Gao AY et al. demonstrated that Pim-1 kinase enhanced the production of IL-6 by activating the activity of NF-κB, thereby promoting the senescence of low-passage IPF fibroblasts [46]. Thus, these data suggest that the sustained activation of NF-κB promotes cellular senescence and aggravates PF by activating SASP-related factors.

### 3.7. Sirtuin Regulatory Pathway

Sirtuins (SIRTs) are a group of nicotinamide dinucleotide-dependent deacetylases that are ubiquitously present in cells and exert a beneficial impact on age-related diseases, including PF. Within the Sirtuin family, mammals possess seven subtypes, namely SIRT1–SIRT7, which primarily impede cellular senescence by preventing telomere attrition, promoting DNA-damage repair, and maintaining genome integrity, [101]. Studies have indicated that the level of SIRT1 was decreased in human lung fibroblasts of replicative senescence and cigarette-extract-induced senescent ATII cells [102,103]. Conversely, the upregulation of SIRT1 ameliorated primary mouse pulmonary fibroblast senescence and PF induced by vitamin D deficiency by suppressing the TGF-β1/IL-11/MEK/ERK signaling pathway [104]. In addition, in PM2.5-exposed mice, the specific deficiency of SIRT3 in AT2 cells significantly increased the expression of senescence-associated proteins, such as p16 and p21, and exacerbated PF by inhibiting P53 deacetylation. Conversely, SIRT3 overexpression can reduce cell senescence of PM2.5-exposed AT2 cells [105]. Moreover, in the TGF-β-induced primary human bronchial epithelial cell senescence model, SIRT6 overexpression inhibited TGF-β-induced cell senescence by increasing p21 degradation [42]. Therefore, targeting SIRTs to eliminate senescent cells or delay senescence may be an effective way to prevent PF.

## 4. Cellular Senescence of AECs and Fibroblasts in the Pathogenesis of Pulmonary Fibrosis

### 4.1. Alveolar Epithelial Cell

The current understanding of PF pathophysiological mechanisms involves recurrent damage to AECs, leading to metabolic dysfunction, senescence, and the abnormal activation and repair of epithelial cells. As progenitor cells of AECs, ATII cells have the functions of synthesizing and secreting pulmonary surfactants, maintaining pulmonary homeostasis and alveolar regeneration. In instances of lung tissue damage, ATII cells, which exhibit a cuboidal shape, can undergo differentiation into thin and flat ATI cells, thereby participating in the repair of alveolar epithelium [106]. However, in recurrent microinjuries, dysfunctional ATII cells not only fail to sustain normal physiological lung regeneration but also interact with mesenchymal cells, immune cells, endothelial cells, and other cell types through various mechanisms to trigger the activation of fibroblasts and myofibroblasts, leading to PF [107]. Single-cell RNA sequencing studies showed that senescent ATII cells were significantly elevated in the population in the lung tissues of PF patients compared with normal lung tissues, and senescence-related signaling pathways such as oxidative stress and p53 signaling were abnormally activated [23,108]. In addition, the PF caused by the SARS-CoV-2 infection in recent years may be due to the senescence of ATII caused by the virus [109,110]. And the elimination of senescent ATII cells and the inhibition of senescence-related signaling pathways contribute to the alleviation of PF [23,111]. ATII cells are extremely sensitive to senescence, and their stem cell potential and repair mechanism gradually fail with senescence. Consequently, ATII cells not only lose their ability to self-renew and proliferate but also impede their differentiation into ATI cells. ATII cells in a partially differentiated or intermediate state can induce senescence-associated differentiation disorders, such as PF [75,112]. In addition, Wu H et al. demonstrated that the impaired differentiation of ATII cells into ATI cells hampers alveolar regeneration. This impairment exposes ATII cells to a sustained mechanical tension, activating the TGF-β signaling pathway and promoting PF [113].

Senescent ATII cells express and secrete SASP-related proteins, including pro-inflammatory factors, pro-fibrotic factors, and tissue-remodeling factors, which play a crucial role in the triggering and progression of PF. A growing body of evidence indicates that ATII cells isolated from the lung tissues of patients with IPF and mice with experimental pulmonary fibrosis exhibit an elevated expression and secretion of SASP factors in vitro [23,108,111]. SASP factors react on senescent cells and neighboring cells through autocrine and paracrine manners, resulting in an impaired immune function and cell dysfunction that leads to a continuous inflammatory response, forming a vicious cycle and thus accelerating the senescence process. For example, Rana T et al. reported that senescent ATII cells activated alveolar macrophages by secreting IL-4 and IL-13, thus participating in the pathogenesis of PF [68]. Furthermore, some genetic variants also mediate PF by regulating SASP secretion in ATII cells. The dysfunction of telomeres in ATII cells exhibits characteristic features of cellular senescence, including the upregulation of cytokines involved in inflammation and immune response, thereby increasing susceptibility to PF [114,115]. In Hermansky–Pudlak syndrome, ATII cells excessively produce monocyte chemotactic protein-1 (MCP1), which leads to the recruitment and activation of alveolar macrophages and enhances the production of TGF-β, thereby promoting fibrotic remodeling [116]. Significantly, the clearance of senescent ATII cells reduces the expression of SASP factors, including IL-6, TNFα, TGF-β, MMP12, and Serpine1, thereby alleviating PF [29,111]. Furthermore, extracellular vesicles (EVs), as a constituent of SASP, facilitate the transfer of various factors such as proteins, lipids, and genetic materials to recipient cells, enabling cell-to-cell communication and influencing the phenotype of neighboring cells. Research has demonstrated that EVs play a crucial role in mediating the impact of senescent cells on their microenvironment [117]. EVs significantly increased in BALF from IPF patients and experimental pulmonary fibrosis models [118]. These EVs from the bronchial epithelial cells of IPF patients transfer senescence to adjacent healthy cells by transporting miRNAs and may cause a positive feedback loop so that the senescence wave circulates in the epithelial cells and promotes the disease state of IPF [119]. Furthermore, EVs secreted by lung fibroblasts from IPF patients accelerate epithelial cell senescence by increasing mitochondrial damage in epithelial cells [120]. Thus, SASP is regarded as a crucial mediator in the modulation of PF by senescent ATII cells. Wound repair following lung tissue injury is crucial for maintaining normal lung homeostasis and function. Normal wound healing involves the recruitment of fibroblasts, the deposition of ECM, and the differentiation of myofibroblasts, which heal the wound by secreting collagen and generating contractile forces. However, uncontrolled repair can lead to excessive ECM production and can induce PF [121]. Studies have shown that senescent AECs can promote the excessive activation of lung fibroblasts by increasing SASP expression during the progression of PF [100,122]. Tian et al. demonstrated that ATII cell senescence driven by PTEN loss led to the overproduction of IL-6 and IL-8, which in turn increased collagen accumulation in human lung fibroblasts [100]. Moreover, Chen X et al. found that the senescence of ATII cells induced the activation of lung fibroblasts by activating the Wnt/β-catenin signaling pathway, ultimately leading to PF [122].

### 4.2. Lung Fibroblasts

Fibroblasts are commonly recognized as the cells that create and maintain the anatomically rich ECM and play a vital role in maintaining the stretching and elastic recoiling of the lung during intact respiration. Fibroblasts can provide the necessary niche and location information for neighboring cells by regulating the biomechanics, microstructure, biochemical cues, and soluble mediators, including cytokines, growth factors, and metabolites in the ECM [123]. In addition, fibroblasts also regulate homeostatic balance during lung injury, repair, and remodeling. However, an increasing number of studies have found that primary fibroblasts isolated from the lung tissues of IPF patients exhibit more features of senescence when compared to age-matched controls. Morphologically, senescent lung fibroblasts display an enlarged and flattened appearance, resembling fibroblasts that have undergone replicative failure [124,125]. IPF lung fibroblasts showed a prolonged doubling time during cell culture in vitro, suggesting a decreased proliferation capacity. In contrast to young non-pathological fibroblasts, senescent fibroblasts in culture do not grow in an ordered parallel geometry, and their growth direction is random. In addition, these senescent lung fibroblasts also increased the levels of senescence-associated β-galactosidase, p21, p53, p16, and SASP [125,126,127].

Hecker L et al. found that persistent PF in aged mice was associated with the accumulation of senescent and anti-apoptotic fibroblasts/myofibroblasts [128]. Targeting senescent lung fibroblasts and/or myofibroblasts with anti-senescence drugs can effectively alleviate bleomycin-induced PF and improve lung function in mice, suggesting that lung fibroblast senescence plays a detrimental role in the development of PF [29,129]. Senescent lung fibroblasts influence the senescence of neighboring cells and the local microenvironment in a paracrine manner by secreting SASP, thereby aggravating the progression of fibrosis. Compared with a blank medium, co-culture with a conditioned medium obtained from senescent lung fibroblasts with AECs resulted in the decreased proliferation of AECs [130]. Kadota T et al. reported that EVs derived from lung fibroblasts isolated from IPF patients increase cellular senescence and pulmonary fibrosis by increasing mitochondrial reactive oxygen species and DNA damage in AECs [120]. In addition, the differentiation of lung fibroblasts into myofibroblasts plays a crucial role in fibrogenesis. A study found that the senescence of fibroblasts contributes to their differentiation into myofibroblasts, thereby promoting PF [76].

On the contrary, some research has shown that fibroblast senescence plays a beneficial mechanism in inhibiting PF. Li Y et al. reported that the inhibition of HAS2 expression induced fibroblast senescence by up-regulating the expression of p27-CDK2-SKP2, thereby alleviating PF [131]. In addition, Cui H et al. demonstrated that miR-34a induces lung fibroblast senescence. Lung fibroblasts from miR-34a-deficient mice treated with bleomycin showed a significantly reduced senescent phenotype but an increased fibrotic response compared with wild-type mice [132]. These studies suggest that the function of fibroblast senescence in the pathogenesis of PF is complex.

## 5. Treatment Strategies towards Cellular Senescence in Pulmonary Fibrosis

In response to induction by a variety of endogenous and exogenous factors, cells undergo senescence. Senescent cells not only lose their normal physiological functions and regeneration ability but also secrete a series of factors that are harmful to surrounding tissues/cells, which in turn drives the progression of age-related diseases. Many tissues in the body are capable of regeneration through the replacement of defective or worn-out cells with new cells, a process that largely relies on stem cells. Stem cells are precursor cells in the body that lack a fixed role and can develop into different types of cells under appropriate conditions. Tissues typically have their reservoir of stem cells to replenish damaged cells. However, this regenerative process becomes less efficient with age [133]. Many of our organs, such as the lungs, are lined with epithelial cells [133]. In damaged normal lungs, ATII cells act as stem cells, increasing ATI cell turnover through transdifferentiation. In contrast, ATII cells isolated from the lung explants of IPF patients exhibit an increased senescent phenotype and impaired colony-forming ability in vitro, indicating ATII stem cell exhaustion. Therefore, pluripotent stem cells from alternative sources, such as embryonic stem cells, induced pluripotent stem cells, and mesenchymal stem cells, have been used for the treatment of pulmonary fibrosis [134,135,136,137]. However, stem cell therapy still poses challenges and limitations due to high treatment costs, ethical concerns, immunocompatibility, and cell survival [138]. On the contrary, the elimination of senescent cells, intervention of senescence-related signaling pathways, and inhibition of SASP secretion have been proposed to treat various age-related diseases, including PF, Alzheimer’s disease, diabetic kidney disease, and osteoporosis [139]. These potential therapeutics are summarized in Table 1.

### 5.1. Targeting Senescent Cells

The combination of dasatinib (an inhibitor of tyrosine kinases) and quercetin (a flavonoid subclass) (D + Q) can ablate senescent cells to alleviate PF. Schafer MJ et al. found that the elimination of senescent cells by D + Q treatment significantly reduced the mRNA expression of p16 in the lung of Ink-Attac transgenic mice with bleomycin-induced PF. At the same time, the degree of fibrosis and lung function of the mice were also improved [29]. Similar results were observed in both the ATII cell Sin3a-LOF mouse model and the fibrotic 3D lung tissue culture model. Treatment with D + Q effectively reversed the senescence of ATII cells and upregulated the expression of epithelial markers by negatively regulating the expression of p21, p16, and p53 [23,111]. In addition, a two-center, open-label pilot study of 13 IPF patients showed that D + Q treatment significantly improved physical function indicators such as 6MWD and 4 m gait speed. However, no significant changes in lung function and biochemical indicators were found [140].

Navitoclax (ABT-263), an inhibitor of the Bcl-2 protein family, selectively kills senescent cells in culture by inducing cell apoptosis. In an IR-induced PF mouse model, ABT-263 treatment reversed PF by selectively killing senescent ATII cells [141]. ABT-263 was also found to kill senescent fibroblasts in p16-3MR mice and IMR-90 cells, thereby inhibiting fibrosis [142,143]. During fibrosis formation, increasing the source of collagen and decreasing the degradation of collagen accelerate the process. As an inhibitor of fibrinolysis, the increase of PAI-1 content contributes to the accumulation of collagen. In addition, PAI-1 as a marker and mediator of senescence was significantly upregulated in fibrotic lungs [144]. Several studies have shown that PAI-1 induces ATII cell senescence by activating the p53–p21 signaling pathway and inhibiting the degradation of p53 [55,145]. Huang WT et al. discovered that the administration of TM5275 (a PAI-1 inhibitor) effectively suppressed AdTGF-β1-induced PF in mice [55]. Furthermore, TM5275 also inhibited TGF-β1-induced senescence in ATII cells and the secretion of SASP, thereby alleviating PF [145]. Oxidative stress is one of the factors driving cell senescence and is thought to contribute to the progression of fibrotic diseases. Therefore, the use of antioxidants is an effective method of inhibition of PF. The NOX1/4 inhibitor, GKT137831, has demonstrated the ability to reverse age-related PF by mitigating fibroblast senescence and enhancing apoptosis sensitivity [128]. At the same time, the expression of senescence markers p21 and p16 in the lung tissues of GKT137831-treated mice were also decreased [146].

A novel cell-permeable peptide antagonist FOXO4-DRI designed by Baar MP et al. has also been used in the study of various aging-related diseases in recent years. This antagonist blocks the interaction between FOXO4 and p53, leading to the exclusion of p53 from the nucleus, which in turn eliminates senescent cells by triggering apoptosis [147]. In IR and bleomycin-induced PF models, FOXO4-DRI alleviates PF by inducing the apoptosis of senescent fibroblasts or myofibroblasts and down-regulating SASP expression [50,148,149].

Additionally, some plant extracts can also alleviate PF by targeting senescent cells. For example, pentoxifylline was found to regulate PAI-1 expression and inhibit lung fibroblast senescence, thereby exerting an anti-fibrotic effect [150,151]. These data suggest that cell senescence promotes the development of PF. The specific elimination of senescent cells using senolytic drugs may be a promising approach for the treatment of PF.

### 5.2. Targeting SASP

Accumulating evidence suggests that senescent cells secrete multiple forms of SASP, which affect a variety of cellular behaviors and promote age-related tissue dysfunction through the autocrine and paracrine pathways. Modulating SASP expression has been shown to have therapeutic potential in various models of PF. The mTOR signaling pathway, known for its role in promoting anabolism and suppressing catabolism, is considered a crucial target for anti-senescence interventions [152]. In vitro studies have demonstrated that rapamycin, an mTOR inhibitor, can inhibit cellular senescence and alleviate radiation- and paraquat-induced PF in mice [153,154]. A further investigation of the anti-senescence mechanism found that rapamycin could inhibit the secretion of SASP and reduce the expression of SASP-related factors, such as IL6 and IL1 [155,156].

TGF-β, as an essential component of SASP, plays a crucial role in promoting the senescence of epithelial cells and fibroblasts and the formation of PF. However, the activation of latent TGF-β requires extracellular enzymatic reactions or mechanical induction. Several studies have identified αv integrin as a pivotal mediator of TGF-β activation in fibrosis [157,158]. The inhibition of αvβ6 alleviated radiation-induced PF [159]. Recently, Decaris ML et al. showed that PLN-74809, a dual αvβ6/αvβ1 integrin inhibitor, reduced collagen deposition in both fibrotic mice and human lung sections [160]. In addition, a 120-participant, randomized, double-blind evaluation of the efficacy and safety of PLN-74809 in IPF patients has been completed, but the results are not yet available (ClinicalTrials.gov Identifier: NCT04396756). Another randomized, double-blind study of the safety and tolerability of TRK-250, a nucleic acid drug targeting TGFβ gene expression, in patients with IPF has also been conducted, and the results have also not yet been shown (ClinicalTrials.gov Identifier: NCT03727802).

In addition to targeting various forms of SASP cytokines with inhibitors, researchers have also utilized specific antibodies, binding proteins, and nanomaterials to treat PF induced by different etiologies. Zhang LM et al. reported that the neutralization of IL-18 using an IL-18-binding protein significantly reduced bleomycin-induced senescence in primary mouse lung fibroblasts [161]. Furthermore, antibodies or nanoparticles targeting IL-11 are thought to alleviate stress-induced premature and senescence-related PF and improve lung function [32,172]. IL-1β is another crucial component of SASP. Guo J et al. showed that neutralizing IL-1β using an IL-1β monoclonal antibody reduced the number of inflammatory cells and the expression of fibrotic mediators such as TNFα and TGFβ in BALF, thereby alleviating silica-induced PF [162].

Moreover, there have been reports on the inhibitory effects of several natural compounds on PF through the down-regulation of SASP expression or secretion. For example, citrus alkaloid extract alleviates pulmonary fibrosis by down-regulating the etoposide-induced secretion of PDGF-β, CTGF, MCP-1, TNF-α, and PDGF-α in senescent fibroblasts [163]. Baicalein was found to attenuate PF by reducing the expression of PAI-1, TNF-α, MMP-10, and MMP-12 in primary lung fibroblasts isolated from mice induced with bleomycin [164]. Similarly, hesperidin alleviated doxorubicin-induced MRC-5 cell senescence and bleomycin-induced mice lung tissue senescence through inhibiting IL-6 expression, thereby attenuating PF [165]. In addition, ginsenoside Rg1 attenuated paraquat-induced PF by alleviating epithelial senescence and reducing the expression of TNF-α, IL-1β, IL-6, and MMP3 [166]. In summary, these studies suggest that SASP secreted by senescent cells promotes the occurrence and development of PF, and targeting SASP may be an effective method for the treatment of PF.

### 5.3. Targeting Senescence Pathways

Cellular senescence is the stress response of normal cells to various stimuli, regulated by multiple signaling pathways. In recent years, there have been more and more studies on the targeted intervention of senescence-related signaling pathways to regulate PF. SIRTs, a class of anti-senescence molecules associated with inflammatory response and oxidative stress, play a crucial role in various age-related tissue fibrosis diseases. Drugs such as SIRT1 activators SRT2104 and SRT1720 effectively alleviate AECII cell senescence and accelerate AECII cell renewal and differentiation [167,168]. Zhou J et al. demonstrated that increasing SIRT1 expression inhibited the senescence of primary lung fibroblasts isolated from vitamin D-deficient mice lung tissues by down-regulating the TGF-β1/IL-11/MEK/ERK signaling pathway and ameliorated PF induced by vitamin D deficiency [104]. Furthermore, baicalein alleviated bleomycin-induced cell senescence and PF by restoring SIRT3 expression [164]. Therefore, targeting the SIRT regulatory pathway may provide a new strategy for the treatment of PF.

NF-κB, another crucial signaling pathway, regulates cell senescence and affects PF mainly by activating SASP-related cytokines. A decreased NF-κB expression or the inhibition of the signaling pathway down-regulated the expression of senescence markers. ACT001, an inhibitor of NF-ĸB, inhibited IL-6 secretion, reduced cell viability, and impaired the differentiation ability of lung fibroblasts isolated from PF patients [169]. Moreover, inhibiting NF-κB activation using BMS-345541 or SR12343 was shown to mitigate cell senescence and enhance the lifespan of mice [100,170]. Similarly, scutellarin, a plant extract, alleviated PF by inhibiting the NF-κB/NLRP3 signaling pathway [99]. These findings indicate that targeting the NF-κB pathway holds promise for PF treatment; however, further clinical trials are necessary to validate these results.

In addition, the inhibition of other signaling pathways that promote cell senescence has also been found to improve PF. For instance, the targeted inhibition of STAT3 activity by STA-21 reduced the oxidant-induced secretion of inflammatory cytokines such as IL-6 in lung fibroblasts. Meanwhile, the expression of the senescence marker p21 and the activity of β-galactosidase also decreased [171]. Moreover, silencing p53 improved Serpine1-induced ATII cell senescence [55]. These findings suggest that attenuating senescence-related signaling pathways holds promise as a therapeutic approach for PF treatment.

## 6. Conclusions

Compelling evidence supports that cellular senescence is a significant driver behind age-related lung diseases, including PF. Senescent AECs and fibroblasts, along with the SASP factors they secrete, collectively serve as both initiators and effectors in driving the advancement of PF. Therefore, various approaches targeting cellular senescence, including the targeted clearance of senescent cells, intervention in aging-related signaling pathways, and the inhibition of SASP secretion, have emerged as potential therapeutic strategies for alleviating PF and improving lung function (Figure 1). However, there is currently no research to confirm the complete elimination of senescent cells by targeted anti-senescence drugs. The issue of drugs targeting cells also needs further consideration. Additionally, the majority of drug studies are focused on animal experiments, lacking reliable preclinical and clinical research. Addressing these issues would contribute to the design and development of anti-senescence drugs and promote their progress in improving PF.

## Figures and Tables

**Figure 1 ijms-24-16410-f001:**
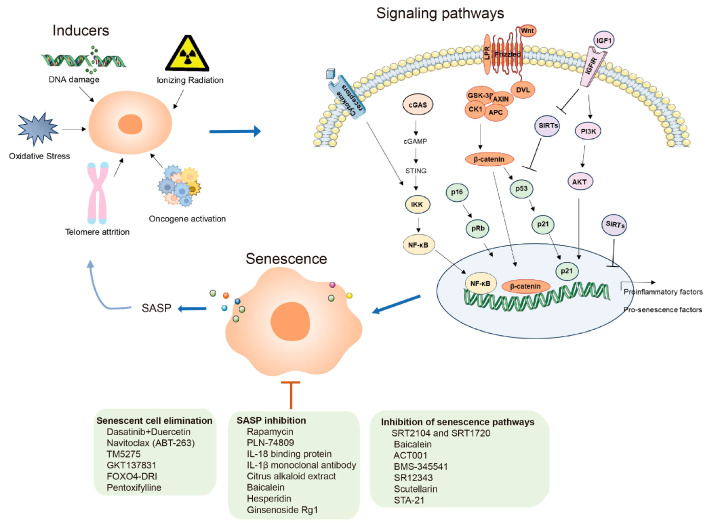
A schematic diagram illustrating the triggers of cellular senescence in pulmonary fibrosis, the signaling pathways involved in its regulation, and therapeutic approaches targeting senescence.

**Table 1 ijms-24-16410-t001:** Treatment strategies towards cellular senescence in pulmonary fibrosis.

Treatment	Proposed Mechanism(s)	Effect	References
Dasatinib + Quercetin	Senolytic (↓p16, ↓p21, ↑apoptosis)	↓Fibrosis in mice ↓Senescence cells and SASP	[23,29,111,140]
Navitoclax (ABT-263)	Bcl2 inhibitor	↓Fibrosis in mice ↓Senescence cells	[141,142,143]
TM5275	PAI-1 inhibitor	↓Fibrosis in mice ↓Senescence cells and SASP	[55,144,145]
GKT137831	NOX1/4 inhibitor	↓Fibrosis in mice ↓Senescence cells	[128,146]
FOXO4-DRI	FOXO4 blocker	↓Fibrosis in mice ↓Senescence cells and SASP	[50,147,148,149]
Pentoxifylline	Senolytic (↓PAI-1, ↑phosphorylated PKA)	↓Fibrosis in mice ↓Senescence cells	[150,151]
Rapamycin	mTOR inhibitor	↓Fibrosis in mice ↓SASP	[152,153,154,155,156]
PLN-74809	Dual αvβ6/αvβ1 integrin inhibitor	↓COL1A1 expression in IPF explanted lung tissue slices	[157,158,159,160]
IL-18 binding protein	Neutralization of IL-18	↓Senescence cells and SASP	[161]
IL-1β monoclonal antibody	Neutralization of IL-1β	↓Fibrosis in mice ↓SASP, modulating the Th1/Th2 balance	[162]
Citrus alkaline extract	Down-regulated the expression of SASP	↓Fibrosis in mice ↓SASP	[163]
Baicalein	Restoration of SIRT3 expression	↓Fibrosis in mice ↓SASP	[164]
Hesperidin	Inhibition of IL6/STAT3 signaling pathway	↓Fibrosis in mice ↓Senescence	[165]
Ginsenoside Rg1	Enhancing autophagy	↓Fibrosis in mice ↓Senescence cells and SASP	[166]
SRT2104, SRT1720	SIRT1 activator	↓Fibrosis in mice ↓p53, ↓p21,	[104,167,168]
ACT001, BMS-345541, SR12343	NF-ĸB inhibitor	↓Fibrosis in mice ↓Senescence	[100,169,170]
Scutellarin	Inhibition of NF-κB/NLRP3 signaling pathway	↓Fibrosis in mice	[99]
STA-21	Inhibiting STAT3 activity	↓Fibrosis in mice ↓Senescence	[171]

## Data Availability

The datasets presented in this study can be found in online repositories. The names of the repository/repositories and accession number(s) can be found in the article.

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
