# Peer review of "Cellular Senescence: A Troy Horse in Pulmonary Fibrosis"

_ijms, 2023, doi:10.3390/ijms242216410_

Round 1
Reviewer 1 Report
Comments and Suggestions for Authors
This is a comprehensive review manuscript focusing on cell senescence in the pathogenesis of pulmonary fibrosis. The topic is important and interested in the field. The manuscript is well written with tons of information. The contents are well organized but they are not well comprehensively summarized. The authors well categorized and listed the causes, pathways, and pathogenesis of cell senescence in PF. If the author could further summarize and present these information with table(s) and/or graph(s) will significantly improve the quality of MS. In addition, it will be better if information revealed from recently scRNA studies in IPF will be great.
1, Although cell senescence was the main topic of this review MS tried to address, an extended discussion of stem cell potential will support their viewpoints;
2, There are several other review articles discussed the same topic have been published in the past years, our understanding in cell senescence in PF is increased with studies using scRNA analysis, the authors better to include some findings from recent scRNA studies in IPF to update our understanding the impotence of senescence in the pathogenesis of this disease;
3, Some references of review articles in this topic and latest scRNA analysis studies should be cited;
4, try to use tables and graphs to comprehensively summarize findings described in the MS, that will help readers to better understand the importance and status of the field"
Author Response
Dear reviewer,
We appreciate the letter and comments concerning improvement to the manuscript entitled "Cellular senescence: a troy horse in pulmonary fibrosis". We have revised the original manuscript carefully correspondingly to the comments and constructive suggestions. To make the changes easily recognizable, all revised portion are marked in yellow background in paper. Please find below the point-by-point responses to the suggestions and comments from you.
We hope the corrections meet your demand. Please let us know if anything else needed.
Many thanks for your time and consideration.
Sincerely,
Guoying Yu PhD, Professor
Director, State Key Laboratory of Cell differentiation and Regulation
Dean, College of Life Science
Henan Normal University
Point-by-Point Responses to the Comments of You
Reviewer 1
Comments and Suggestions for Authors
This is a comprehensive review manuscript focusing on cell senescence in the pathogenesis of pulmonary fibrosis. The topic is important and interested in the field. The manuscript is well written with tons of information. The contents are well organized but they are not well comprehensively summarized. The authors well categorized and listed the causes, pathways, and pathogenesis of cell senescence in PF. If the author could further summarize and present these information with table(s) and/or graph(s) will significantly improve the quality of MS. In addition, it will be better if information revealed from recently scRNA studies in IPF will be great.
1, Although cell senescence was the main topic of this review MS tried to address, an extended discussion of stem cell potential will support their viewpoints;
Response: We appreciate your comment and have incorporated the research on stem cell potential into the revised manuscript. Thank you very much for your reminder.
2, There are several other review articles discussed the same topic have been published in the past years, our understanding in cell senescence in PF is increased with studies using scRNA analysis, the authors better to include some findings from recent scRNA studies in IPF to update our understanding the impotence of senescence in the pathogenesis of this disease;
Response: Thank you for your helpful suggestion, and we have incorporated some studies on scRNA analysis in IPF into the revised manuscript.
3, Some references of review articles in this topic and latest scRNA analysis studies should be cited;
Response: Thank you very much for your suggestion, and we have added the corresponding references in the revised manuscript.
4, try to use tables and graphs to comprehensively summarize findings described in the MS, that will help readers to better understand the importance and status of the field"
Response: Thanks for your consideration to this point and we have included a table outlining a potential therapeutic strategy for cellular senescence in the revised manuscript.
Many thanks are given to you for your valuable suggestions.
We hope that the revised manuscript is now suitable for publication.
Reviewer 2 Report
Comments and Suggestions for Authors
This is a very nice, well-written, and very comprehensive review of cell senescence in pulmonary fibrosis. It will be of great value to the field.
The authors need to address two things that are puzzling:
1) Fibrosis in the lungs involves a large increase in tissue mass that replaces the air spaces. I am puzzled how senescence (cessation of cell proliferation) can co-exist with the remarkable increase in tissue mass.
2) the authors describe how a large number of the major signalling pathways in cells all lead to senescence. But these pathways are used for many things; how do they not activate senescence in other cells?
Minor points
Line
46 ‘.. differentiation of AECs as alveolar..’ I’m confused, do you mean ‘..differentiation of AECs from alveolar..’?
54 myofibroblast of scar formation - - > myofibroblasts and scar formation
55-56 how does growth arrest lead to persistent fibrosis?
108 mediated please be more specific – increased? decreased?
126 a vita inducer - - > an inducer
Fig 1 left green box eliminate - - > elimination
156 – 157 usually Ras activation causes growth, please explain
217 please explain ‘Arctiin-encapsulated DSPE-PEG’
368 is an effective - - > may be an effective
548 human seminal lung sections - - > human lung sections
Author Response
Dear reviewer,
We appreciate the letter and comments concerning improvement to the manuscript entitled "Cellular senescence: a troy horse in pulmonary fibrosis". We have revised the original manuscript carefully correspondingly to the comments and constructive suggestions. To make the changes easily recognizable, all revised portion are marked in yellow background in paper. Please find below the point-by-point responses to the suggestions and comments from you.
We hope the corrections meet with your demand. Please let us know if anything else needed.
Many thanks for your time and consideration.
Sincerely,
Guoying Yu PhD, Professor
Director, State Key Laboratory of Cell differentiation and Regulation
Dean, College of Life Science
Henan Normal University
Point-by-Point Responses to the Comments of You
Reviewer 2
This is a very nice, well-written, and very comprehensive review of cell senescence in pulmonary fibrosis. It will be of great value to the field.
The authors need to address two things that are puzzling:
1) Fibrosis in the lungs involves a large increase in tissue mass that replaces the air spaces. I am puzzled how senescence (cessation of cell proliferation) can co-exist with the remarkable increase in tissue mass.
Response: On the one hand, cellular senescence is different from apoptosis or necrosis, as it represents a state of "permanent" cell cycle arrest. Senescent cells, although they cease to divide, remain metabolically active and exhibit increased size and SA-β-galactosidase activity. Therefore, even in the state of cellular senescence, these cells still occupy air spaces and are not cleared by the body. On the other hand, during the process of pulmonary fibrosis formation, activated lung fibroblasts secrete excessive collagen and other extracellular matrix components, leading to increased lung tissue fibrosis and mass. Although senescent cells are cell cycle arrested, they still maintain metabolic activity and secrete various fibrogenic cytokines and growth factors collectively known as "senescence-associated secretory phenotype", which affect local tissue physiology through autocrine and paracrine mechanisms and contribute to the development of age-related diseases. For instance, senescent alveolar epithelial cells overexpress and secrete excessive Tgfβ, which acts in a paracrine manner on lung fibroblasts, thereby promoting their activation and excessive extracellular matrix secretion, ultimately replacing the alveolar air spaces.
2) the authors describe how a large number of the major signaling pathways in cells all lead to senescence. But these pathways are used for many things; how do they not activate senescence in other cells?
Response: Yes, these signals do indeed participate in other cellular physiological activities. We believe that under normal physiological conditions, these signaling pathways are maintained in a dynamic equilibrium state, coordinating and constraining each other, without being abnormally activated, thus causing changes in cellular physiological functions. However, when excessive endogenous and exogenous factors stimulate the system, many signaling pathways associated with disease occurrence can be abnormally activated or deactivated. Pulmonary fibrosis is the result of repeated damage and abnormal repair of alveolar epithelial cells caused by various risk factors. Due to the abnormal repair process, the dynamic equilibrium between signaling pathways is disrupted. For example, the WNT/β-catenin signaling pathway is primarily involved in cell differentiation regulation and tissue homeostasis. However, Lehmann et al. observed an elevated WNT/β-catenin activity in ATII cells isolated from aged mice compared to those from young mice. Chronic activation of canonical WNT/β-catenin signaling induced a pronounced cellular senescence phenotype in primary ATII cells and MLE-12 cells [1]. Moreover, Kadota T et al. reported that human bronchial epithelial-derived extracellular vesicles alleviated TGF-β-induced myofibroblast differentiation and alveolar epithelial cell senescence by inhibiting WNT signaling pathways, thereby attenuating bleomycin-induced PF in mice [2]. Meanwhile, the induction of senescence in AECs by mitomycin was observed to be regulated through the enhancement of GSK3β phosphorylation [3]. Consequently, the aberrant activation of WNT/β-catenin signaling can promote lung cell senescence by inducing cell cycle arrest, thereby contributing to the progression of PF.
[1] Lehmann M, et al. Chronic WNT/β-catenin signaling induces cellular senescence in lung epithelial cells. Cell Signal. 2020 Jun;70:109588.
[2] Kadota T, et al. Human bronchial epithelial cell-derived extracellular vesicle therapy for pulmonary fibrosis via inhibition of TGF-β-WNT crosstalk. J Extracell Vesicles. 2021 Aug;10(10):e12124.
[3] Xu X, et al. Mitomycin induces alveolar epithelial cell senescence by down-regulating GSK3β signaling. Toxicol Lett. 2021 Nov 1;352:61-69.
Minor points
Line
46 ‘.. differentiation of AECs as alveolar..’ I’m confused, do you mean ‘..differentiation of AECs from alveolar..’?
Response: Thank you very much for your pointing out. The intended meaning of this sentence is as follows. “Physiologically, alveolar epithelial type II (ATII) cells, serving as progenitor cells of the alveoli, differentiate into ATI cells in response to injury. Utilizing organoid culture, single-cell transcriptomics, and lineage tracing, it has been discovered that ATII cells differentiate into ATI cells and acquire a transitional state known as pre-alveolar type 1 cell during the process of maturation. This transitional state exhibits regulation by TP53 signaling, making it susceptible to DNA damage and undergoing transient senescence.” We have rephrased it in the revised manuscript to make it easier for readers to understand.
54 myofibroblast of scar formation - - > myofibroblasts and scar formation
Response: Thank you for your reminder and we have changed it according to your suggestion.
55-56 how does growth arrest lead to persistent fibrosis?
Response: Thank you very much for bringing this to our attention. It was a misrepresentation on our part, and we have rephrased this sentence in the revised manuscript.
108 mediated please be more specific – increased? decreased?
Response: In the context of the paper, "mediated" refers to "increased". We have changed it in the revised manuscript. Thank you for your helpful suggestion.
126 a vita inducer - - > an inducer
Response: Thank you very much for your suggestion, and we have changed it in the revised manuscript.
Fig 1 left green box eliminate - - > elimination
Response: Thank you for your helpful suggestion, and we have changed it in the revised figure 1.
156 – 157 usually Ras activation causes growth, please explain
Response: Although RAS signaling is crucial for generating mitogenic signals that promote cell proliferation, unrestricted RAS activation in primary mammalian cells typically triggers a cascade of molecular and cellular events leading to cellular senescence, a state of permanent cell cycle arrest in which cells remain metabolically active [1]. Overactive RAS promotes cellular senescence through multiple complementary pathways, which may vary depending on the cellular context. It is universally accepted that oncogenic RAS triggers cellular senescence by activating the p53 and p16INK4-Rb pathways, promoting degradation of pro-proliferative proteins, and activating DNA damage response [2-6]. Additionally, increased oncogenic RAS also exacerbates the malignant paracrine activity of senescence-associated secretory phenotype (SASP), thereby enhancing senescence [7].
[1] Serrano M, et al. Oncogenic ras provokes premature cell senescence associated with accumulation of p53 and p16INK4a. Cell. 1997 Mar 7;88(5):593-602.
[2] Di Micco R, et al. Oncogene-induced senescence is a DNA damage response triggered by DNA hyper-replication. Nature. 2006 Nov 30;444(7119):638-42.
[3] Bartkova J, et al. Oncogene-induced senescence is part of the tumorigenesis barrier imposed by DNA damage checkpoints. Nature. 2006 Nov 30;444(7119):633-7.
[4] Mallette FA, et al. The DNA damage signaling pathway is a critical mediator of oncogene-induced senescence. Genes Dev. 2007 Jan 1;21(1):43-8.
[5] Deschênes-Simard X, et al. Tumor suppressor activity of the ERK/MAPK pathway by promoting selective protein degradation. Genes Dev. 2013 Apr 15;27(8):900-15.
[6] Villot R, et al. ZNF768 links oncogenic RAS to cellular senescence. Nat Commun. 2021 Aug 17;12(1):4841.
[7] Coppé JP, et al. Senescence-associated secretory phenotypes reveal cell-nonautonomous functions of oncogenic RAS and the p53 tumor suppressor. PLoS Biol. 2008 Dec 2;6(12):2853-68.
217 please explain ‘Arctiin-encapsulated DSPE-PEG’
Response: The term "Arctiin-encapsulated DSPE-PEG" refers to the bubble-like nanoparticles formed by encapsulating Arctiin using 1,2-distearoyl-sn-glycero-3-phosphoethanolamine and polyethylene glycol 2000 as carriers. We have provided a clear explanation of this in the revised manuscript.
368 is an effective - - > may be an effective
Response: Thank you very much for your suggestion, and we have changed it in the revised manuscript.
548 human seminal lung sections - - > human lung sections
Response: Thank you very much for your suggestion, and we have changed it in the revised manuscript.
Many thanks are given to you for your valuable suggestions.
We hope that the revised manuscript is now suitable for publication.
Round 2
Reviewer 1 Report
Comments and Suggestions for Authors
The authors addressed all my concerns. There is no more concern.